# Biomechanical Efficacy of Three Methods for the Fixation of Posterior Malleolar Fractures: A Three-Dimensional Finite Element Study

**DOI:** 10.3390/diagnostics13233520

**Published:** 2023-11-24

**Authors:** Vincenzo Giordano, Márcio Antônio Babinski, Anderson Freitas, Robinson Esteves Pires, Felipe Serrão de Souza, Luiz Paulo Giorgetta de Faria, Pedro José Labronici, Alexandre Godoy-Santos

**Affiliations:** 1Serviço de Ortopedia e Traumatologia Prof. Nova Monteiro, Hospital Municipal Miguel Couto, Rua Mário Ribeiro 117, Rio de Janeiro 22430-160, Brazil; felipeserrao@yahoo.com.br (F.S.d.S.); lpgiorgetta@gmail.com (L.P.G.d.F.); 2Clínica São Vicente, Rede D’or São Luiz, R. João Borges 204, Rio de Janeiro 22451-100, Brazil; 3Departamento de Morfologia, Universidade Federal Fluminense, Avenida Prof. Hernani-Mello 101, Niterói 24210-150, Brazil; mababinski@gmail.com; 4HOME—Hospital Ortopédico e Medicina Especializada, Quadra 613—Conjunto C—Asa Sul, Brasília 70200-730, Brazil; andfreitas28@gmail.com; 5Departamento do Aparelho Locomotor, Universidade Federal de Minas Gerais (UFMG), Avenida Prof. Alfredo Balena 190, Belo Horizonte 30130-100, Brazil; robinsonestevespires@gmail.com; 6Hospital Felício Rocho, Avenida do Contorno 9530, Belo Horizonte 30110-934, Brazil; 7Departamento de Ortopedia e Traumatologia, Universidade Federal Fluminense (UFF), Avenida Marquês do Paraná 303, Niterói 24220-000, Brazil; labronicipedro@gmail.com; 8Serviço de Ortopedia e Traumatologia Prof. Dr. Donato D’Ângelo, Hospital Santa Teresa, Rua Paulino Afonso 477, Petrópolis 25680-003, Brazil; 9Faculdade de Medicina, Universidade de Sao Paulo, Rua Dr. Ovídio Pires de Campos, São Paulo 05403-010, Brazil; alexandrelemegodoy@gmail.com; 10Hospital Israelita Albert Einstein, Avenida Albert Einstein 627, São Paulo 05652-900, Brazil

**Keywords:** ankle fracture, posterior malleolus fracture, syndesmosis, finite element method, biomechanical study

## Abstract

**Introduction:** We investigated the biomechanical behaviour of different fixations of the tibial posterior malleolus (TPM), simulating distinct situations of involvement of the tibiotalar articular surface (TTAS) through a finite element model (FEM). **Material and methods:** A 3D computer-aided design model of the left ankle was obtained. The materials used were divided according to their characteristics into ductile and non-ductile, and all materials were assumed to be linear elastic, isotropic, and homogenous. Three different fracture lines of the TPM were defined, with sagittal angles of 10°, 25°, and 45°. For biomechanical comparison, different constructions using a trans-syndesmotic screw (TSS) only (Group T), a one-third tubular plate only with (Group PT) and without (Group PS) a TSS, and a locked compression plate with (Group LCPT) and without (Group LCPS) a TSS were tested. FEM was used to simulate the boundary conditions of vertical loading. Load application regions were selected in the direction of the 700 N Z-axis, 90% on the tibia and 10% on the fibula. Data on the displacement and stress in the FEM were collected, including the total principal maximum (MaxT) and total principal minimum (MinT) for non-ductile materials, total displacement (desT), localized displacement at the fragment (desL), localized displacement at syndesmosis (desS), and Von Mises equivalent stress for ductile materials. The data were analysed using ANOVA and multiple comparison LSD tests were used. **Results:** For TPM fractures with sagittal angles 10° and 25°, desL in the PT and LCP groups was significantly lower, as well as Von Mises stress in Group LCPT in 10°, and PT and LCPT groups in 25°. For TPM fractures with a sagittal angle of 45°, desL in the LCP group and Von Mises stress in Group LCPS and LCPT were significantly lower. We found that any TPM fracture may indicate instability of the distal tibiofibular syndesmosis, even when the fragment is small. **Conclusion:** Our study showed that in fragments involving 10% of the TTAS, the use of a TSS is sufficient, but when the involvement is greater than 25% of the TTAS, either a non-locked or locked plate must be used to buttress the TPM. In posterior fragments affecting 45% or more of the TTAS, the use of a locking plate is recommended.

## 1. Introduction

Ankle fractures are among the most common skeletal bone injuries, with an estimated prevalence of distal tibiofibular syndesmosis injury in up to 11% of cases [1]. At least one third of these fractures affect the tibial posterior malleolus (TPM) (Volkmann’s fragment), which seems to have a direct impact on worsening clinical outcomes. A recent biomechanical study using a cadaveric model showed that, in a situation of absence of load, neither the injury to the syndesmotic complex nor the fracture TPM significantly influenced the position of the fibula in the fibular notch [2]. However, this study was based on a designed surgery without additional injuries in a weight-bearing free situation. In addition, the simulated injury during this cadaver study differed in mode from the injuries usually seen after trauma. Thus, considering current knowledge, most ligament injuries of distal tibiofibular syndesmosis and/or TPM fracture will require surgical stabilization [3].

Despite this, there is still no consensus on the size of the Volkmann fragment that potentially generates instability of the syndesmotic complex, and it remains unclear what is the best approach for this fracture, especially regarding when and how its reduction and fixation take place. Currently, the most accepted indication is that fractures of the TPM should be operated when they affect at least 20% of the tibiotalar articular surface (TTAS) or present articular diastasis ≥2.0 mm [4]. In these cases, the reduction is preferably carried out directly with fixation using an anti-glide plate. In another cadaveric study, it was shown that the rigidity of the distal tibiofibular syndesmosis was restored to 70% after the fixation of the TPM and to 40% after the stabilization of the syndesmosis when compared with intact specimens [5].

More recently, Mansur et al. [6] compared the biomechanical behaviour of four different methods used for the fixation of the TPM using a finite element model (FEM), concluding that the use of two 3.5 mm cannulated screws from posterior to anterior provided better fixation resistance in this type of lesion. Despite these findings, these authors did not investigate the role of TPM fixation in fragments smaller than 30% of the TTAS, nor did they use a comparative model with a trans-syndesmotic screw (TSS).

The null hypothesis is that the fixation of the TPM with an anti-glide plate, alone or in association with the use of a TSS, regardless of the size of the fragment and the involvement of the TTAS, is superior to the stabilization performed with a syndesmotic screw alone. Herein, we aimed to biomechanically evaluate the behaviour of different fixations of the TPM, simulating distinct situations of involvement of the TTAS through a FEM.

## 2. Materials and Methods

### 2.1. Finite Element Models of Ankle Joint

The study was approved by the institutional review board of the hospital and did not involve animals nor humans. A 3D computer-aided design (CAD) model of the left ankle was obtained based on fourth generation composite tibia and fibula models. The tibia model had a length of 405 ± 1 mm, a distal tibial joint width of 58 ± 1 mm, and an inner canal of 10 mm in diameter (#3402, Sawbones, Seattle, WA, USA), whereas the fibula model had a length of 384 ± 1 mm, a distal width of 19 ± 1 mm, and an inner canal of 2.5 mm in diameter (#3427-1, Sawbones, Seattle, WA, USA).

The materials used were divided according to their characteristics into ductile (metallic implants) and non-ductile (bone and ligaments). All materials were assumed to be linear elastic, isotropic, and homogenous. The elastic modulus and Poisson’s ratio were obtained from previous studies [6,7,8] (Table 1).

### 2.2. BioCad Preparation

From the synthetic bone models and their syntheses, computed tomography (CT) scan images of the left ankle in neutral unloaded position were obtained and archived in the communication protocol that includes DICOM (Digital Imaging and Communications in Medicine) files. We used the Emotion CT scan with 16 channels (Siemens™, Munich, Germany), with a resolution of 512 × 512, and a slice interval of 1.0 mm. The DICOM file was imported to the program InVesalius™, for the three-dimensional (3D) reconstruction of the anatomical structure. Based on a set of two-dimensional images obtained through CT equipment, the program allows the generation of 3D virtual models of the regions of interest. After reconstructing the DICOM images in 3D, the program allows the generation of 3D files in stereo lithography (STL) format.

The 3D virtual models of each system (bone and implants) were made by the program Rhinoceros™ 6 (Robert McNeel and Associates, Seattle, WA, USA). The geometric models of cartilage and ligament were built according to the anatomical relationship between the bones and ligaments following the methodology described by Guan et al. [9]. Ligaments included anterior tibiofibular ligament, posterior tibiofibular ligament, anterior talofibular ligament, posterior talofibular ligament, deltoid ligament, and calcaneofibular ligament, and the thickness of the cartilage was approximately 1.0 mm (Figure 1). The analysis by FEM was performed in the program SimLab™ (HyperWorks, Troy, CA, USA), using the Optistruct solver, in a computer with Intel Xeon (Santa Clara, CA, USA) processor (CPU E-3-1240 v3 3.40 GHz, with 32 GB RAM in a 64-bit Windows 7 operating system).

The horizontal projection of the distal articular surface of the tibia was taken as the reference plane. The AB projection line in the plane of the fibular notch and parallel to the medial fibula cortex was taken as a reference line and the point of intersection of the posterior ankle and medial ankle was point “O” [9]. The 1/4 point of the AB line was taken as point C. The intersection point of the posterior ankle and medial ankle was the “O” point, and the OC line was connected as the horizontal fracture line (Figure 2).

The angle between the fracture line and the *z*-axis on the sagittal reconstruction images was defined as the sagittal angle of the posterior malleolus fracture (Figure 3). Taking the *z*-axis in the sagittal reconstruction images, three different fracture lines of the posterior malleolus were defined, with sagittal angles of 10°, 25°, and 45°. The sagittal angles were defined to simulate a fracture line considered stable (10°), a fracture line that affects the fibular notch (25°), and a fracture line that affects a large part of the tibiotalar loading area (45°), representing a tibial pylon. All reference points and lines were established by the main author (VG), and were subsequently reviewed by two scholars, one of the authors (AF), and the engineer who carried out all the finite element analysis.

The implants used were a 2.7 mm, straight, five-hole, titanium alloy locked compression plate (X49.681), a 3.5 mm non-locked titanium alloy 1/3 tubular plate, four holes (245–401), and a 50 mm 3.5 mm titanium alloy cortical screw. All implants were formatted as indicated by the manufacturer’s dimensional characteristics (DePuy Synthes—J and J Company, Raynham, MA, USA). Two lag screws were inserted through the distal holes of both plates to apply interfragmentary compression to the posterior malleolus. Three locked screws were used in the proximal holes of the 2.7 mm locked compression plate. The transsyndesmotic screw was inserted as the positioning screw to hold the syndesmosis in place.

### 2.3. File Conversion

In the InVesalius™ program, all slices were imported to obtain the STL file with the images that would be used in the process of obtaining the 3D solid. This allowed for multiplanar generation, which made it possible to evaluate images in the sagittal, coronal, and axial planes, and volume. From the volume, the creation of the 3D surface was carried out, allowing the selection of the regions of interest using masks and/or filters. This allowed the file to be hidden or portrayed, according to the algorithm in question, generating the 3D surface.

### 2.4. Simulation and Boundary Conditions

The FEM was used for the stability simulations of the different assemblies. First, the files were imported into the Simlab™ program, with identification of each part of the digital models. After controlling the meshes of each part, care was taken to always maintain the size of the element, so that there were no contact problems between the different parts in the simulations. A tetrahedral element was adopted to form the meshes.

To define the boundary conditions, load application regions were selected in the direction of the 700 N *z*-axis, 90% on the tibia and 10% on the fibula. On the *x*- and *y*-axes, no loading was applied. Subsequently, the movement restriction regions (fixed) were delimited, marked in all directions of the displacement and rotation *x*-, *y*-, and *z*-axes. These restrictions ensured that the system had perfect alignment without displacement and/or rotation. The friction coefficient of the fracture surface was 0.3. Ligaments were applied to the model as preload by reducing the length of ligaments by 2% with zero load.

### 2.5. Load Application and Fixation Site

After controlling the meshes of each part, care was taken to always maintain the size of the element, so that there are no contact problems between the different parts (ankle and synthesis) in the simulations (Figure 4).

For biomechanical comparison, each of three different fracture lines of the posterior malleolus (10°, 25°, and 45°) were tested with the following constructions: a transsyndesmotic screw only (Group T), a one-third tubular plate only (Group PS), a one-third tubular plate plus a transsyndesmotic screw (Group PT), a locked compression plate only (Group LCPS), and a locked compression plate plus a transsyndesmotic screw (Group LCPT). We also tested the models without fixation (Group S) to evaluate the degree of instability generated by the different fracture lines.

Data on the displacement and stress in the FEM were collected, including the total principal maximum (tension, MaxT) and total principal minimum (compression, MinT) for non-ductile materials, total displacement (desT), localized displacement at the fragment (desL), localized displacement at syndesmosis (desS), and Von Mises equivalent stress for ductile materials. Because of the mechanical properties, the (-) sign represents the direction of compressive stress (MinT). All results are presented in absolute values and percentiles between the models.

### 2.6. Statistical Analysis

Descriptive statistics and percentages were used to determine means and differences. One-way ANOVA and multiple comparison LSD tests were used to determine the mean difference. The *p*-value was considered significant if *p* < 0.05. The statistical software SPSS 21.0 (SPSS Inc., Chicago, IL, USA) was used.

## 3. Results

The number of nodes was defined as shown in Table 2. All results are presented for the three different fracture lines of the posterior malleolus (10°, 25°, and 45°) and the tested constructions are summarized in Table 3, Table 4 and Table 5.

For posterior malleolus fractures with a sagittal angle of 10°, there was no statistical significance between groups T, PT, and LCPT for desT and desS (*p* < 0.05); however, desL was significantly lower in the PT and LCPT groups, although there was no statistically significant difference between them. There was statistically significant reduction for the Von Mises stress in Group LCPT (*p* < 0.0001), which denotes less demand on implants in this construction (Table 3). Figure 5 illustrates the MaxT for Group LCPS.

For posterior malleolus fractures with a sagittal angle of 25°, there was no statistical significance between groups T, PT, and LCPT for desT and desS (*p* < 0.05); however, desL was significantly lower in the PT and LCPT groups, although there was no statistically significant difference between them. There was a statistically significant reduction in Von Mises stress in the PT (*p* < 0.0001) and LCPT (*p* < 0.0001) groups, which denotes a lower requirement for implants in these constructions (Table 4).

For posterior malleolus fractures with a sagittal angle of 45°, there was no statistical significance between groups T, PT, and LCPT for desT and desS (*p* < 0.05); however. desL was significantly lower in Group LCPT. There was a statistically significant reduction in Von Mises stress in group LCPS (*p* < 0.05) and LCPT (*p* < 0.05), although there was no statistically significant difference between them, which denotes a lower requirement for implants in these constructions (Table 5 and Figure 6).

## 4. Discussion

Although recent studies have raised the question of the role of the posterior malleolus in ankle stability, there is still a great debate as to whether, when, and how posterior malleolus fractures should be repaired [4,5,6,10,11]. It has been reported that fractures of the posterior malleolus occur in approximately 46% of Weber type B or C ankle fracture–dislocations and have a close relationship to the injury or instability of the distal tibiofibular syndesmosis, especially the posterior inferior tibiofibular ligament (PITFL). Of relevance, it was shown that the PITFL is the most resistant in syndesmosis stability, with the majority of LTFPI injuries occurring in the form of delamination of the posterior malleolus [12,13].

In a large cohort of rotationally unstable ankle fractures without posterior malleolus fractures, Warner et al. [13] observed that accurate and stable syndesmotic reduction is a significant component of restoring the ankle mortise after unstable ankle fractures. They found that in 122 ankle fractures, the PITFL was delaminated from the posterior malleolus in 97% (119/122) of cases, with a smaller proportion (3%; 3/122) having an intrasubstance PITFL rupture. Despite this, until recently, the classic indications for surgical treatment were the presence of a fragment >25% and posterior instability of the ankle [4].

Here, we report the biomechanical behaviour of different fixations of the posterior tibial malleolus, simulating different situations of involvement of the tibiotalar articular surface through a finite element model. In addition to presenting our results, we aimed to define the role of combined osteosynthesis of the Volkmann fragment using a posterior buttress plate and transsyndesmotic screw. Our findings showed that there is no need for direct reduction of a posterior malleolar fracture involving less than 10% of the articular surface, but it is necessary to stabilize the distal tibiofibular joint with a transsyndesmotic screw. When the posterior fragment of the tibia involves 25% of the articular joint, we find that it is necessary to use a buttress plate, but there is no difference if a locked or non-locked plate is used. The addition of a transsyndesmotic screw to the posterior plate does not significantly reduce the total maximum displacement or Von Mises stress. Finally, for the posterior malleolus involving 45% of the articular surface, the use of a locking plate significantly reduced the maximum total displacement and Von Mises stress, with or without the addition of a transsyndesmotic screw.

Our findings reinforce what was demonstrated by Bartoníček et al. [14], that the involvement of the fibular notch due to a posterior malleolus joint fracture potentially generates distal tibiofibular joint incongruity, leading to instability and post-traumatic osteoarthritis of the ankle. Although the magnitude of syndesmotic malreduction that can lead to inferior patient-reported outcomes remains unclear, theoretically both rotational and sagittal translation of the fibula may occur after posterior malleolus non-reduction, malreduction, or insufficient fixation, and may go undetected using traditional imaging methods [15,16].

In a retrospective cohort study of 87 patients with complete syndesmosis injury evaluated with radiographs and CT scan, Andersen et al. [17] observed a syndesmosis malreduction rate of 32%. These authors noted that a difference of 2.0 mm in the anterior distal tibiofibular relationship predicts unsatisfactory clinical results, with 79% specificity and 61% sensitivity. In another study, Sagi et al. [18] found a similar malreduction rate, as high as 44%, in patients in whom the syndesmosis underwent closed reduction.

Indeed, it has been reported that the overall functional outcome of ankle fractures with posterior malleolar involvement are significantly worse compared with uni- or bi-malleolar fractures [4], which is probably due to the unrecognized or occult injury to the PITFL. Biomechanical studies have shown a significant decrease in the tibiotalar contact area as the size of the posterior malleolar fragment exceeds 33%, a significant increase in posterior subluxation between 25% and 40%, and an increase in stress on the remainder of the tibiotalar joint [19,20,21,22,23]. In this context, adequate stabilization of the distal tibiofibular syndesmosis is essential for the normal movement of the ankle joint, necessary for weight transmission and walking, and the posterior malleolus is of great importance for this.

Mason et al. [11,23] recommended the use of one transsyndesmotic screw for extra-articular posterior malleolar fractures, sustained by avulsion from the distal posterior tibial cortex by the pull of the PITFL, and open reduction and internal fixation for posterolateral fractures of the tibia extending into the incisura fibularis, regardless of the size of the posterior malleolar fragment. Our findings indicate the same; however, based on FEM results, we suggest the use of a locked plate with or without a transsyndesmotic screw for posterior malleolar fragments greater than 45% of the tibiotalar articular joint.

This study has some limitations. As this is an experimental biomechanical study using FEM, limitations inherent to the project include the lack of correlation of mechanical findings with the expected biological response during tissue healing in a malleolar fracture of the ankle. In addition, we used normal-density fourth generation composite tibia and fibula models; therefore, our findings cannot be extrapolated to situations in which bone stock is not adequate, such as elderly patients with osteoporotic or insufficiency fracture. Also, the load application was performed in a single positioning of the ankle joint. Since this joint can undergo several load changes in the most varied gait cycles, it seems necessary to validate our findings simulating other ankle positions. Finally, our study evaluated a transsyndesmotic screw versus buttress plate constructs. We did not evaluate the mechanical behaviour of other implants currently used for the treatment of distal tibiofibular syndesmosis injury, such as elastic fixation devices and suture tape. Furthermore, we did not evaluate torque control during bony insertion for both the syndesmosis fixation screw and the cortical screws used in the plates. In fact, this is a critical factor in the biomechanical stability of implants, as screws are often insufficiently or excessively tightened, which can lead to unnecessary or compromised assembly. Moldovan and Bățagă [24] recommended the use of a digital screwdriver that allows the measurement of torque during insertion, depending on bone density and the type of screw used.

However, our study has strengths. First, the use of synthetic bone models in biomechanical experiments has been shown to increase the potential for high biomechanical fidelity, low variability across specimens, decreased financial burden, and ease of use compared with human cadaveric bones [25]. Second, FEM may be applied to almost any orthopaedic problem which is related to a biomechanical issue, representing the most used computational technique in orthopaedic research [25]. Third, we used highly controlled and reproducible testing conditions, simulating three different situations of posterior malleolus fracture, in which there continues to be controversy about the need to stabilize the distal tibiofibular syndesmosis and how to do it. Finally, in addition to the bone structures, in our model, the ligamentous structures of the distal tibiofibular syndesmosis were considered in the computational model, which almost perfectly replicate the physiological condition of this joint. It has been suggested that the creation of an anatomical 3D FEM of the ankle joint is necessary for the realistic prediction of load transfer and stress distribution for preclinical analysis of numerous constructs and implants [26].

## 5. Conclusions

In conclusion, any fracture occurring in the posterior malleolus of the ankle can be indicative of potential instability within the distal tibiofibular syndesmosis, even when the fragment is relatively small in size. Our research findings have demonstrated that when the fracture involves approximately 10% of the tibiotalar articular surface, employing a transsyndesmotic screw is generally effective. However, in cases where the involvement exceeds 25% of the articular surface, it becomes necessary to consider the utilization of either a non-locked or locked plate to provide reinforcement to the posterior malleolus. For posterior fragments that impact 45% or more of the articular surface, it is strongly recommended to apply a locking plate as the preferred method of treatment.

## Figures and Tables

**Figure 1 diagnostics-13-03520-f001:**
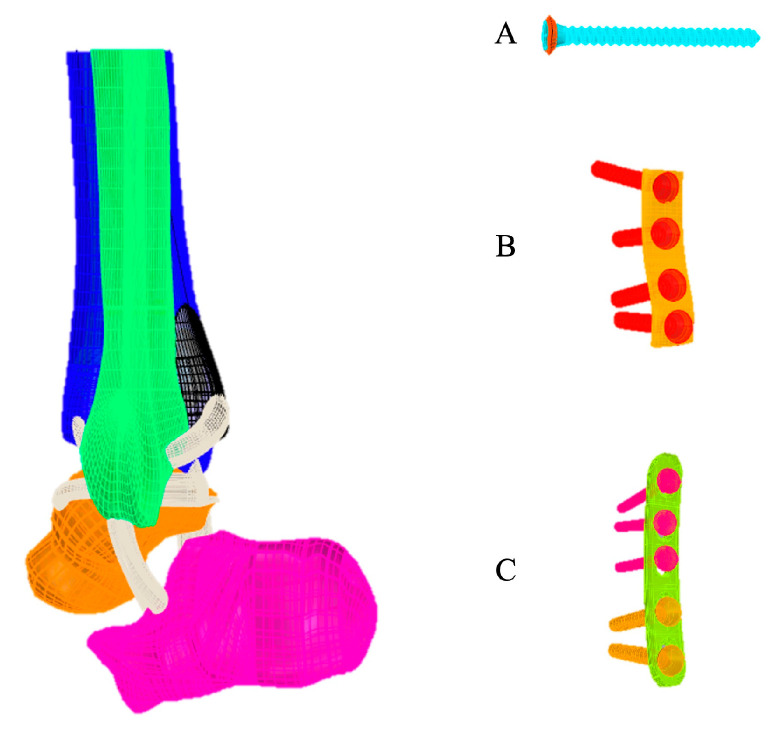
Three-dimensional files in STL format: (**A**) cortical screw 3.5 mm, tricortical, with 30° anterior inclination, 35 mm from the tibiotalar articular surface; (**B**) non-locking 1/3 tubular plate 3.5 mm, four cortical screws 3.5 mm; (**C**) locking plate 2.7 mm, straight, five holes, with two cortical screws 2.7 mm and three locking screws 2.7 mm.

**Figure 2 diagnostics-13-03520-f002:**
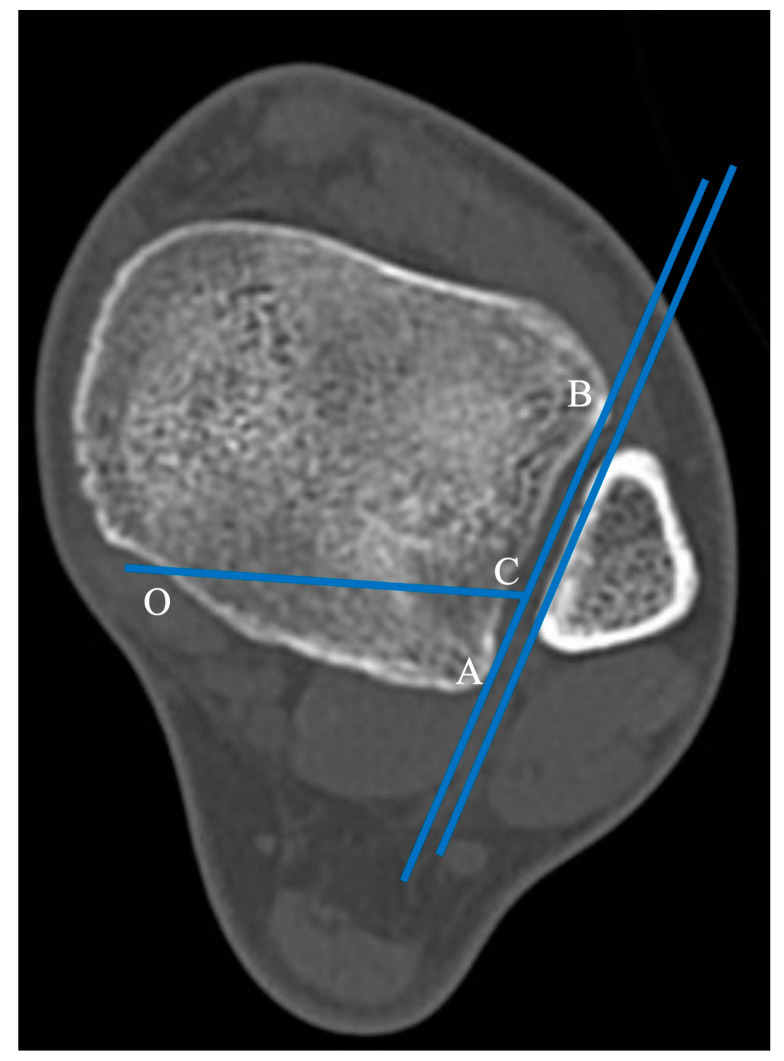
Axial cut of a normal distal tibia and fibula. The AB projection line in the plane of the fibular notch and parallel to the medial fibula cortex was taken as a reference line and the point of intersection of the posterior ankle and medial ankle was point “O”. The 1/4 point of the AB line was taken as point C. The intersection point of the posterior ankle and medial ankle was the “O” point, and the OC line was connected as the horizontal fracture line.

**Figure 3 diagnostics-13-03520-f003:**
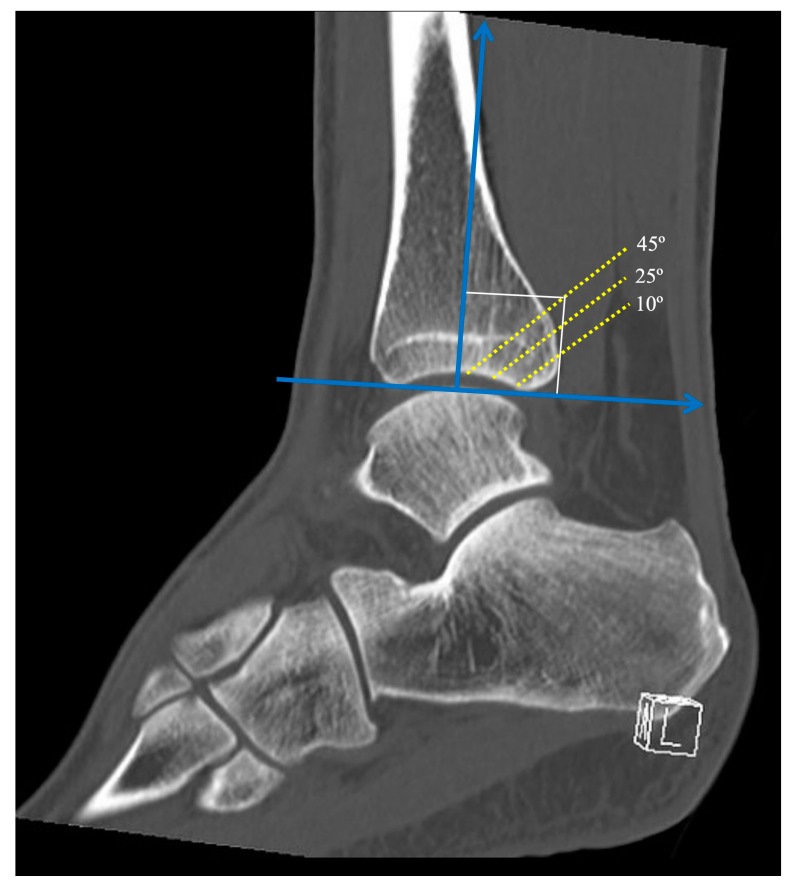
The angle between the fracture line and the *z*-axis on the sagittal reconstruction images was defined as the sagittal angle of the posterior malleolus fracture. Note the three different sagittal angles with fragments involving 10°, 25°, and 45° of the articular surface.

**Figure 4 diagnostics-13-03520-f004:**
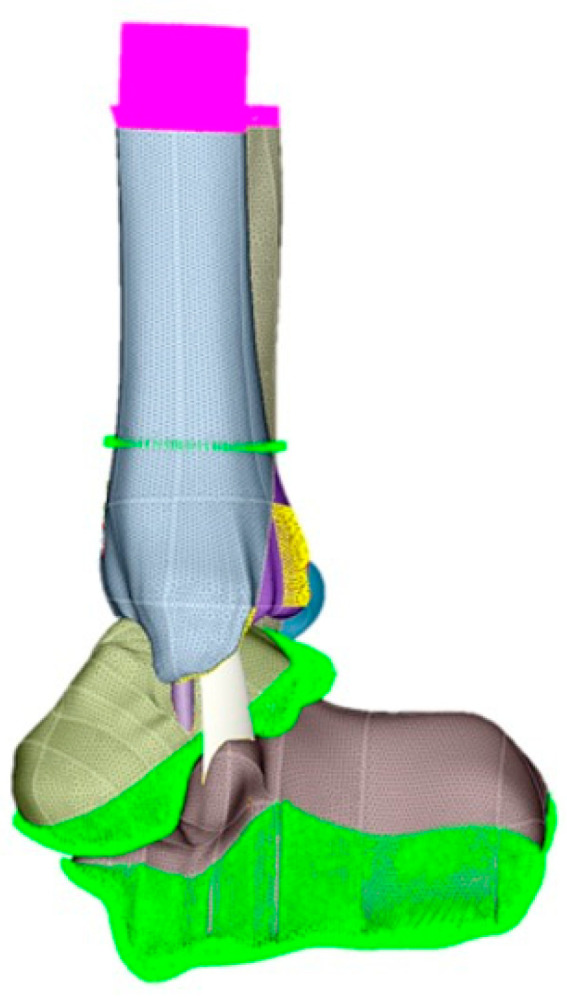
During simulations, the size of the element was always maintained. A tetrahedral element was adopted to form the meshes.

**Figure 5 diagnostics-13-03520-f005:**
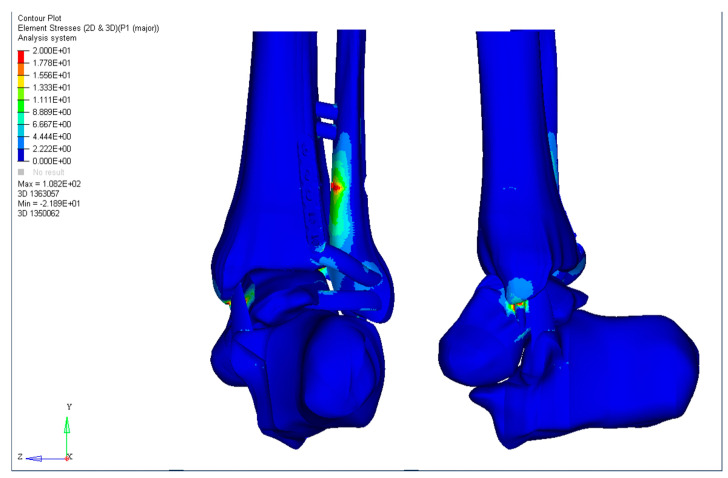
MaxT for Group LCPS for posterior malleolus fractures with a sagittal angle of 10°.

**Figure 6 diagnostics-13-03520-f006:**
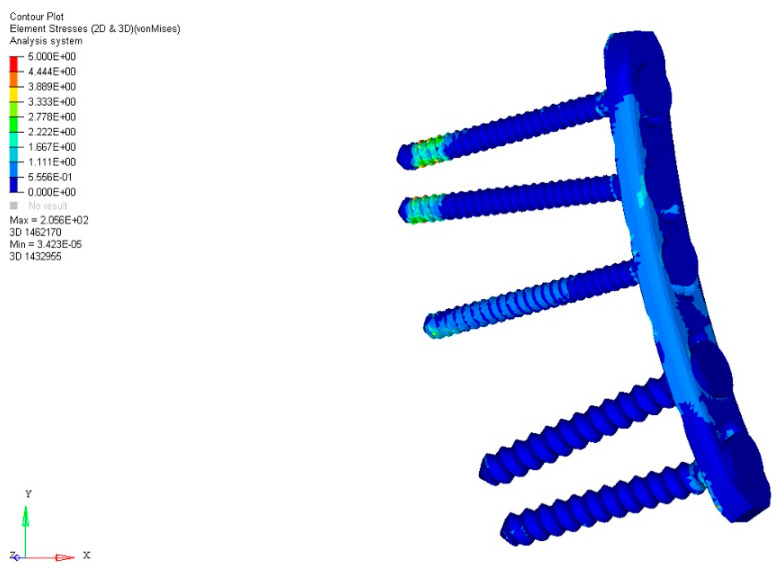
There was a statistically significant reduction in Von Mises stress in the locked compression plate groups for posterior malleolus fractures with a sagittal angle of 45°. Note the slight increase in von Mises stresses at the tip of the two most proximal locked screws, although there are no signs that the material would yield or fracture.

**Table 1 diagnostics-13-03520-t001:** Material properties.

Material	Properties	
	Modulus of Elasticity (Mpa)	Poisson’s Ratio (v)
Cortical bone	17,000	0.30
Trabecular bone	477	0.30
Titanium alloy	19,300	0.30
Ligaments	260	0.49

Source: Serviço de Ortopedia e Traumatologia Prof. Nova Monteiro–Hospital Municipal Miguel Couto, Rio de Janeiro, Brazil.

**Table 2 diagnostics-13-03520-t002:** Number of nodes according to the element.

Assembly	Elements	Nodes
1	914931	1,472,169
2	917697	1,411,642
3	951306	1,531,426
4	965302	1,620,877
5	971306	1,731,426
6	985302	1,820,877

Source: Serviço de Ortopedia e Traumatologia Prof. Nova Monteiro–Hospital Municipal Miguel Couto, Rio de Janeiro, Brazil.

**Table 3 diagnostics-13-03520-t003:** Data on the displacement and stress in the FEM for posterior malleolus fractures with a sagittal angle of 10°.

	Group S	Group T	Group PS	Group PT	Group LCPS	Group LCPT
MaxT	138.9	158.9	84.4	171.6	108.2	107.6
MinT	−111.5	−221.5	−78.74	−153.2	−71.03	−138.4
desT	18.34	13.34	13.73	13.15	13.67	13.63
desL	4.7	3.1	1.3	1.1	0.8	0.6
desS	6.4	4.1	7.1	3.6	7.0	3.4
Von Mises	n/a	2533	1296	1290	1056	654.6

Source: Serviço de Ortopedia e Traumatologia Prof. Nova Monteiro–Hospital Municipal Miguel Couto, Rio de Janeiro, Brazil. Legends: MaxT—total principal maximum; MinT—total principal minimum; desT—total displacement; desL—localized displacement at the fragment; desS—localized displacement at syndesmosis; Group S—no fixation; Group T—transsyndesmotic screw only; Group PS—one-third tubular plate only; Group PT—one-third tubular plate plus a transsyndesmotic screw; Group LCPS—locked compression plate only; Group LCPT—locked compression plate plus a transsyndesmotic screw; n/a—not available.

**Table 4 diagnostics-13-03520-t004:** Data on the displacement and stress in the FEM for posterior malleolus fractures with a sagittal angle of 25°.

	Group S	Group T	Group PS	Group PT	Group LCPS	Group LCPT
MaxT	122.3	149.0	101.2	143.5	113.6	117.6
MinT	−97.34	−186.0	−87.47	−112.2	−128.4	−108.4
desT	20.18	13.67	13.72	13.40	10.60	14.63
desL	5.1	3.4	1.7	1.3	1.0	0.7
desS	6.7	4.3	7.2	3.5	7.1	3.5
Von Mises	n/a	2445	831.2	739.3	1248	694.8

Source: Serviço de Ortopedia e Traumatologia Prof. Nova Monteiro–Hospital Municipal Miguel Couto, Rio de Janeiro, Brazil. Legends: MaxT—total principal maximum; MinT—total principal minimum; desT—total displacement; desL—localized displacement at the fragment; desS—localized displacement at syndesmosis; Group S—no fixation; Group T—transsyndesmotic screw only; Group PS—one-third tubular plate only; Group PT—one-third tubular plate plus a transsyndesmotic screw; Group LCPS—locked compression plate only; Group LCPT—locked compression plate plus a transsyndesmotic screw; n/a—not available.

**Table 5 diagnostics-13-03520-t005:** Data on the displacement and stress in the FEM for posterior malleolus fractures with a sagittal angle of 45°.

	Group S	Group T	Group PS	Group PT	Group LCPS	Group LCPT
MaxT	90.04	103.3	112.8	127.9	122.0	127.8
MinT	−78.87	−142.3	−78.87	−126.1	−82.57	−100.4
desT	24.17	15.34	15.55	15.23	20.81	19.88
desL	5.4	3.6	2.9	2.2	1.2	0.7
desS	7.0	4.6	7.3	3.8	7.2	3.6
Von Mises	n/a	1145	1256	1119	594.8	794.8

Source: Serviço de Ortopedia e Traumatologia Prof. Nova Monteiro–Hospital Municipal Miguel Couto, Rio de Janeiro, Brazil. Legends: MaxT—total principal maximum; MinT—total principal minimum; desT—total displacement; desL—localized displacement at the fragment; desS—localized displacement at syndesmosis; Group S—no fixation; Group T—transsyndesmotic screw only; Group PS—one-third tubular plate only; Group PT—one-third tubular plate plus a transsyndesmotic screw; Group LCPS—locked compression plate only; Group LCPT—locked compression plate plus a transsyndesmotic screw; n/a—not available.

## Data Availability

The data presented in this study are available on request from the corresponding author. The data are not publicly available due to authorship protection.

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
