# Peer review of "Biomechanical Efficacy of Three Methods for the Fixation of Posterior Malleolar Fractures: A Three-Dimensional Finite Element Study"

_diagnostics, 2023, doi:10.3390/diagnostics13233520_

Round 1
Reviewer 1 Report
Comments and Suggestions for Authors
This is an interesting paper with appropriate study design. I only have a few comments to improve the paper.
A few reference points and lines were established. I wonder how many observers were there? If there is a single observer, how did the authors ensure intra-rater reliability?
Dicom files from real patients were used. Please explain why ethical approval was not necessary.
Comments on the Quality of English LanguageNo comment.
Author Response
Dear Reviewer, thanks for your comments.
Follow our responses:
Comment 1: This is an interesting paper with appropriate study design. I only have a few comments to improve the paper.
Response: Thanks again. We responded all.
Comment 2: A few reference points and lines were established. I wonder how many observers were there? If there is a single observer, how did the authors ensure intra-rater reliability?
Response: Thank you for your comment. All reference points and lines were established by the main author (VG), and were subsequently reviewed by two scholars, one of the authors (AF) and the engineer who carried out all finite element analysis. We believe this has increased intra-rater reliability. This information was added to the text so that readers can understand how this process was carried out.
Comment 3: Dicom files from real patients were used. Please explain why ethical approval was not necessary.
Response: DICOM files were obtained from a 3D computer-aided design model of the left ankle based on 4th generation composite tibia and fibula models (Lines 98 and 99). No human exams were used in the study, therefore an informed consent form or the approval by an ethics committee was necessary.
Reviewer 2 Report
Comments and Suggestions for Authors
The research aim is to analyse and compare 3 types of osteosynthesis for posterior tibial plafond fractures using FEA. The subject is interesting and modern.
The abstract is structured appropriately.
The introduction transposes the research into the topic and formulates the objective of the study at the end.
In the methodology section the stages of the research are presented. Editing recommendation – the subsections should be in italics and numbered by 2.1; 2.2 etc.
For the results section there some concerns. The tables should be edited in APA style. As a general rule for reporting the p values: if p value is greater than 0.05 should be reported with two decimal values, if p value is between 0.001 and 0.05 should be reported with three decimal places and if values shown on output as 0.000 should be reported as <0.0001; please explain the p <0.00001 (lines 242; 262 etc.); p.0.005 (line 259; line 274 etc.) and p <0.05 (line 276 etc.) or make the necessary corrections. There seems also that 2 fonts have been used in this section.
The discussions interpret the research results and relate them to other scientific papers. However, information about the importance of torque control of the screws in relation to osteosynthesis failure could also be addressed in relation to Flaviu Moldovan, Tiberiu Bățagă. Torque Control during Bone Insertion of Cortical Screws. https://doi.org/10.1016/j.promfg.2020.03.070.
The conclusions are concise and clear.
The references are adequate and properly edited but can be extended as suggested above
Author Response
Dear Reviewer, many thanks for your positive comments and suggestions. Follow our responses:
Comment 1: The research aim is to analyse and compare 3 types of osteosynthesis for posterior tibial plafond fractures using FEA. The subject is interesting and modern. The abstract is structured appropriately. The introduction transposes the research into the topic and formulates the objective of the study at the end. In the methodology section the stages of the research are presented. Editing recommendation – the subsections should be in italics and numbered by 2.1; 2.2 etc.
Response: Thanks again for your positive comments.
Comment 2: For the results section there some concerns. The tables should be edited in APA style. As a general rule for reporting the p values: if p value is greater than 0.05 should be reported with two decimal values, if p value is between 0.001 and 0.05 should be reported with three decimal places and if values shown on output as 0.000 should be reported as <0.0001; please explain the p <0.00001 (lines 242; 262 etc.); p.0.005 (line 259; line 274 etc.) and p <0.05 (line 276 etc.) or make the necessary corrections. There seems also that 2 fonts have been used in this section.
Response: Thanks again. Regarding the tables, we just followed the template suggested by the editor. However, we will ask the editor to check that this is appropriate and in line with the recommendations to authors given by MDPI. Regarding the presentation of the p values, apologies for our mistake when writing the results. All p values were corrected according to your recommendations. Finally, regarding the font used in this section, we used Palatino Linotype normal. The only difference is that for the main text it was used size 10 and for the tables size 9.
Comment 3: The discussions interpret the research results and relate them to other scientific papers. However, information about the importance of torque control of the screws in relation to osteosynthesis failure could also be addressed in relation to Flaviu Moldovan, Tiberiu Bățagă. Torque Control during Bone Insertion of Cortical Screws. https://doi.org/10.1016/j.promfg.2020.03.070.
Response: Thanks for suggesting the study published by Moldovan and Bățagă. This was added to the text and references.
Comment 4: The conclusions are concise and clear. The references are adequate and properly edited but can be extended as suggested above
Response: Once again, thanks for your suggestions and comments.
Reviewer 3 Report
Comments and Suggestions for Authors
Comments to the manuscript entitled Biomechanical efficacy of three methods for the fixation of posterior malleolar fractures: a three-dimensional finite element study. This is interesting research that have been realized very well. The English grammar is fantastic, and the entire article is great research. Minor changes are required.
Abstract: Authors must structure the abstract describing section as introduction, methods, results and conclusion.
Keywords. Ok
Introduction. Is a bit short but the aim of the study, objectives and hypothesis.
Material and methods. Are well describe. Nevertheless, I would like to know if any statistical outcomes were developed in the research and if any statistically software was used.
Discussion. A great discussion that includes a limitation and strength section. Congratulations
Conclusion. Explain all the research according to the aim of the study.
Author Response
Dear Reviewer, many thanks for the positive comments. Follow our responses:
Comment 1: Comments to the manuscript entitled Biomechanical efficacy of three methods for the fixation of posterior malleolar fractures: a three-dimensional finite element study. This is interesting research that have been realized very well. The English grammar is fantastic, and the entire article is great research. Minor changes are required.
Response: Many thanks.
Comment 2: Abstract: Authors must structure the abstract describing section as introduction, methods, results and conclusion.
Response: Abstract was structured according to your suggestion.
Comment 3: Keywords. Ok. Introduction. Is a bit short but the aim of the study, objectives and hypothesis.
Response: Thanks again.
Comment 4: Material and methods. Are well describe. Nevertheless, I would like to know if any statistical outcomes were developed in the research and if any statistically software was used.
Response: We used the statistical software SPSS 21.0 (SPSS Inc., Chicago IL) was used. This is mentioned in lines 231 and 232.
Comment 5: Discussion. A great discussion that includes a limitation and strength section. Congratulations. Conclusion. Explain all the research according to the aim of the study.
Response: We appreciate your positive and kind comments. Thanks a lot.